# The composition of cosmic rays according to the data on EAS cores

S. B. Shaulov[1*], V. A. Ryabov[1], S. E. Pyatovsky[1], A. L. Shepetov[1] and V. V. Zhukov[2]

**1** P. N. Lebedev physical institute of Russian academy of science, Moscow, Russia
**2** Tien-Shan High Mountain Science Station of LPI, Almaty, Kazakhstan

⋆ ser101@inbox.ru

*21st International Symposium on Very High Energy Cosmic Ray Interactions
(ISVHECRI 2022)
Online, 23-28 May 2022*

## Abstract

The main purpose of this work is to find the causes of the break of the cosmic ray spectrum at an energy of 3 PeV, which is called the knee. The solution of the problem is associated with the determination of the nuclear composition of cosmic rays in the knee area. The conclusions of this work are based on the analysis of the characteristics of the EAS cores obtained using X-ray emulsion chambers. According to these data, a number of anomalous effects are observed in the knee region, such as scaling violation in the spectra of secondary hadrons, an excess of muons in EAS with gamma families and others. At the same energies equivalent to 1-100 PeV the laboratory system colliders show scaling behavior. So analysis of the data on the EAS cores suggests that the knee in their spectrum is formed by a component of cosmic rays of a non-nuclear nature, possibly consisting of stable (quasi-stable) particles of hypothetical strange quark matter, which named strangelets. This is the only model of the knee compatible with the magnetic rigidity of the nuclear spectra break R=100 TV. In fact, stranglets are stable heavy quasi-nuclei with a positive electric charge of Z=30-1000, so the mechanism of their acceleration coincides with the nuclear one. The break of the cosmic ray spectrum can be associated with a significantly larger mass of strangelets compared to nuclei.

## 1 Introduction

The question on the nature of the sharp change of power index at the energy of $(3-4)$ PeV, named knee, in the energy spectrum of cosmic rays (CR) is one of the main problems of modern astrophysics. The only way to solve it is determination of the mass composition of cosmic rays in the knee area. There exist two methods suitable for the purpose: investigation of

the electromagnetic component of extensive air showers (EAS), and the study their cores. In spite of its higher higher difficulty, the latter method is significantly more sensitive, since the energetic hadrons of an EAS which carry most of the information on the properties of primary cosmic ray particle are concentrated in the core. In the current paper we summarize the results of a study of EAS cores made at high altitudes in the atmosphere with the use of X-ray emulsion chambers.

## 2 Magnetic rigidity of the knee region

According to modern data, the acceleration of CR nuclei occurs on the shock waves left after the supernova explosion. This is an electromagnetic process, so the maximum acceleration energy of nuclei with different charges Z should be determined by the same value of magnetic rigidity $R = E_0/Z$.

So one of the main issues for us is the magnetic rigidity of the knees in the individual spectra of different cosmic ray nuclei. This values can be deduced from the cut-off energy of the proton spectrum. Such data were obtained in the AS$\gamma$ experiment (Tibet, 4300 m above sea level) [1] using the emulsion chambers and burst detectors which were accompanying with air shower array. It was obtained that the slope of the observed spectrum is steeper for primary energies $E_0 > 10^{14}$ eV than that extrapolated from the lower energy region, irrespective of the interaction models assumed in the simulation.

The figure 1 at [1] shows the results obtained based on the simulation code CORSIKA with interaction models of QGSJET01 and SIBYLL 2.1.

At energies above 100 TeV, the slope of the proton spectrum equal to -3.01 ± 0.11 and −3.05 ± 0.12 for the spectra obtained using the QGSJET and SIBYLL models, respectively, which are steeper than that with slope -2.74 ± 0.01 in the energy range below 100 TeV.

Also, in the figure 2 in the same work the energy dependence of the relative part of heavy cosmic ray nuclei is shown. An accelerated growth of the fraction of nuclei heavier than helium just in the knee region seen at this plot confirms the conclusion about the value of the cut-off of proton spectrum at the rigidity $R_p \sim 0.1$ PV.

The resulting value of magnetic rigidity is thirty times different from the generally accepted value of 3 PV. Therefore in the same figures the results of AS$\gamma$ experiment are compared with those by the KASCADE experiment [2,3]. In the case of using the SIBYLL and QGSJET models, the data of the KASCADE experiment obtained from the analysis of only the electromagnetic component of the EAS at sea level differ several times, while the AS$\gamma$ data for the same models coincide.

At the same time, the KASCADE data is in better agreement with the AS$\gamma$ data when using the SIBYLL model than when using the QGSJET. However, the conclusion about the magnetic rigidity $R_{max} = 3$ PV was obtained precisely when using the QGSJET model.

As previously noted, the conclusions based on the analysis of only the electromagnetic component of the SHAL largely depend on the interaction model used, whereas when taking into account the data on the trunks of the SHAL, this dependence is significantly less. Therefore, the conclusions of the AS$\gamma$ experiment seem to be more reliable.

However, in this case, a problem arises, because the spectrum of nuclei ends with an iron group at an energy of 3 PeV. Nevertheless, we intend to show that there is a way out. With a magnetic rigidity of $R_{max} = 0.1$ PV, further CR spectrum at energies above 3 PeV can be built, assuming the presence of a non-nuclear component in the CR. Next, we show that there are experimental prerequisites for such a hypothesis.

## 3 Non-nuclear component in CR

The hybrid experiment HADRON was carried out at Tien-Shan at an altitude of 3330 meters above sea level. The HADRON installation combined a network of scintillation detectors for registering extensive air showers (EAS), an X-ray emulsion chamber (XREC) with an area of $162\,\mathrm{m}^2$ for registering EAS cores and an underground muon hodoscope at a depth of 20 meters of water equivalent for registering muons with an energy of $E_\mu \geq 5$ GeV.

The combination of EAS detectors and XREC in the same experimental installation made it for the first time possible to trace the dependence of the spectra of the most energetic hadrons in EAS cores on the total number of charged particles $N_e$ which defines the primary energy $E_0$ of the shower [5,6].

As it has revealed, the integral spectra of individual hadrons in EAS core were of an exponential form, $N_h(\geq x_h) \sim x_h^{-\beta}$, where $x_h = E_h/E_0$ is a dimensionless normalized value for the hadron energy $E_h$. The dependence of the slope index $\beta$ of these spectra on the mean values $N_e$ (and $E_0$) of corresponding EAS is shown in [5,6]. As it is seen there, the hadron spectra keep nearly constant slope around $-1.9$ until the shower size values $N_e \sim 10^{6.1}$ ($E_0 \sim 3$ PeV), and then the slope diminishes considerably up to value -1.2, i.e. the spectra become harder and that is the energy of hadrons increases.

Thus, in the area of $N_e = 10^6 - 10^{7.7}$ ($E_0 = 3 - 200$ PeV), there is a scaling violation in the hadron spectrum. The difference from scaling behavior is $4.5\sigma$.

On the other hand, in special experiments on the LHCf and RHICf colliders, [7–11] it was shown that scaling should be observed at energies equivalent to 1-100 PeV in a laboratory system. These data, as well as the quantum chromodynamics, are an evidence of the absence of any major change in characteristics of nuclear interactions at considered energies $E_0$.

Thus, it can be stated that in the reactions caused by cosmic ray particles it was detected a sharp violation of scaling in the energy spectra of secondary hadrons which can be neither observed in collisions of accelerated protons and ions nor predicted by simulations based on the modern models of hadron interaction. In cosmic rays, the slope of energy spectra of secondaries decreases, i.e. the average energy of the produced hadrons increases, within the range of interaction energies $E_0 \simeq (3 - 200)$ PeV. As it follows from accelerator data, the observed scaling violation in cosmic rays cannot be explained by any change in the characteristics of nuclear interaction. The only remaining interpretation possibility of the scaling violations of secondaries spectra in EAS cores is an appearance at $E_0 \simeq 3$ PeV of some highly penetrating component in the total flux of primary cosmic rays which reduce the loss of energy by the development of particles cascade in the atmosphere. Therefore, the violation of scaling properties observed in the interaction of cosmic ray particles should be associated with a change of their initial composition.

It could be possible to explain the increase of the average energy of secondary hadrons at $E_0 \simeq 3$ PeV by appearance of a new proton component among the cosmic rays in this region. However, such an interpretation attempt contradicts to the data of the HADRON experiment on the muon content of EAS cores which are shown in [12].

According to these data, in those EAS, in which the high-energy hadrons where detected by the imprint of their interaction in the XREC, the number of muons $N_\mu$ in average is *higher* then the mean $N_\mu$ value calculated over all detected showers without any pre-selection, and this difference starts to be visible just in the region of the knee at $E_0 \simeq 3$ PeV.

An increase in the energy of hadrons and the number of muons in the same events contradicts any model of the nuclear composition of CR. The increase in energy can be attributed to the appearance of the proton component. However, in this case, the energy dissipation in the atmosphere should decrease and the number of muons in the EAS should also decrease, which is in direct contradiction with the situation [12].

Seemingly, the only remaining option is to suppose that some non-nuclear component does appear in the knee area of the primary cosmic rays spectrum. This assumption can be associated with possible presence among the cosmic rays of the hypothetical particles of the strange quark matter.

## 4 The model

To be capable to reach the Earth, all the particles of cosmic rays have to be stable. Of course, such are the atomic nuclei constituted by the nucleons, which, in turn, consist of the $u$ and $d$ quarks, as it was stated by Zweig and Gell-Mann [13–15]. However, under certain conditions the systems which include the $s$ quark as well, but have much higher baryon number than that of the usual nuclei can become stable. Witten showed that the stable quark matter in the form of quark stars can exist in the nature [16].

When destroyed, such stars can emit into the surrounding space the heavy multi-quark globules consisting of the $u$, $d$, and $s$ quarks, and having the atomic weight $A \sim (10^3 - 10^8)$ and the positive electric charge $Z$ from about $\sim 30$ up to thousands. Such particles, named *strangelets*, are essentially quasi-nuclei. Just like nuclei, strangelets can be accelerated by the shock waves.

In the frames of the strangelet hypothesis, we can propose the following composition model of the cosmic rays within the knee region [17].

## 5 Conclusion

We suggest a model represented in figure 10 of [17]. According to the model, the composition of cosmic rays remains to be of nuclear kind up to the energies of $E_0 \simeq 3$ PeV. From the knee and up to the ankle the cosmic rays predominantly consist of strangelets. Above the ankle, in the range of extragalactic cosmic rays, the strangelets may be also present, but this is a special question. The change in the slope of the spectrum at the 3 PeV knee is caused by appearance of the non-nuclear component of cosmic rays, which is much heavier than the nuclear component.

## Acknowledgements

We express our gratitude to all participants of the HADRON experiment, who ensured the long-term operation of the installation in difficult conditions of the highlands.

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
