# Peer review of "The composition of cosmic rays according to the data on EAS cores"

_SciPost Physics Proceedings, doi:SciPost Phys. Proc. 13, 014 (2023)_

## Round 3 · Referee Report · Anonymous (Referee 1) · 2022-9-16

Strengths

1) This paper is based on the data of the HADRON experiment, that measured the hadronic EAS component together with electromagnetic and muonic ones.

Weaknesses

1) The main weakness of this paper is that the knee interpretation is based only on the 2006 Tibet-ASgamma results. Few data from other experiments are cited and only in preliminary and very old versions. No data published after 2006 are cited, while many important results about the knee of the cosmic spectrum have been published. Therefore the model proposed doesn't take into account many other results that can change the hypothesis on which the paper is based. 2) The paper is difficult to read as many results are cited from other pubblications (one of those only in Russian). But I understand that this can be due to the fact that the paper is conference proceeding with a limit in the page number.

Report

This is interpretation of the knee is a very peculiar one based on hypothesis that ignore results from other experiment (i.e. ARGO, KASCADE, KASCADE-Grande, ICE-TOP).
The hypothesis of new cosmic ray stranglets components seems ad hoc for these results and it look strange that such component only shows its influence above the knee, not being visible at lower energies.
In my opinion such interpretation is not convincing and should be better explained (but maybe this is not possible in a short pubblication like a proceedings, if this is the case I invite the authors to submit a longer article to a journal).

Requested changes

I suggest few changes either of content or only in the wording. - page 2 line 6. "....higher complicity". Maybe you can change "complicity" with a more appropriate one. - page 2 line 9. "In the current message...." message --> contribution or paper - Section 2 line 4. stifness --> rigidity - Section 2 4th paragraph. Please check "equils" - Section 2 5th paragraph. "figer 2" --> "figure 2" - Section 2 end of 5th paragraph. Rp=0.1 PeV --> Rp=0.1 PV (I suppose) - SIBILL --> the name of the model is SIBYLL - page 3 1st paragraph. "Obviously, a more .....". To me is not obvious at all the analysis of Tibet-ASg is more reliable than the others. I would use either a less hard sentence or better justify this claim. -page 3 end of 1st paragraph. "There is a "desert"". Again is not so clear to me why there is this "desert", there are other models that can explain the cosmic radiation above the knee. This is true in a peculiar data interpretation. - page 3 First line of section 3. Is true that Tien Shan is at 3330 m a.s.l., I remember that it is above 5000 m. Please check. -page 3 4th paragraph. In this paragraph (beginning with "Exceptional hardness...". Could you please try to explain more clearly why the "spectra reveal...... non scaling behaviour"- - page 4 4th paragraph. "....means preferable origination...." please check this sentence, I think it can be written in a better way. - page 4 last two lines. "....the following composition model". Is this model the one shown in figure 10 of reference 17? If yes I think that it can dicrectly adressed in this way in this line. If not I do not understand the sentence.

  • validity: low
  • significance: low
  • originality: ok
  • clarity: low
  • formatting: reasonable
  • grammar: below threshold

---

## Editorial Decision

published